# Migrant mobility flows characterized with digital data

**Mattia Mazzoli** [1]*, **Boris Diechtiareff**[2], **Antònia Tugores**[1], **Willian Wives**[2], **Natalia Adler**[3], **Pere Colet**[1], **José J. Ramasco**[1]*

**1** Instituto de Física Interdisciplinar y Sistemas Complejos IFISC (CSIC-UIB), Campus UIB, Palma de Mallorca, Spain, **2** United Nations Children's Fund -UNICEF Brasília, DF, Brazil, **3** United Nations Children's Fund -UNICEF 3 UN Plaza, New York, NY, United States of America

\* mattia@ifisc.uib-csic.es (MM); jramasco@ifisc.uib-csic.es (JJR)

**Data Availability Statement:** The manuscript contains all the information needed to download the data and to replicate the analysis.

**Funding:** MM is funded by the Conselleria d'Innovaci\'o, Recerca i Turisme of the Government

## Abstract

Monitoring migration flows is crucial to respond to humanitarian crisis and to design efficient policies. This information usually comes from surveys and border controls, but timely accessibility and methodological concerns reduce its usefulness. Here, we propose a method to detect migration flows worldwide using geolocated Twitter data. We focus on the migration crisis in Venezuela and show that the calculated flows are consistent with official statistics at country level. Our method is versatile and far-reaching, as it can be used to study different features of migration as preferred routes, settlement areas, mobility through several countries, spatial integration in cities, etc. It provides finer geographical and temporal resolutions, allowing the exploration of issues not contemplated in official records. It is our hope that these new sources of information can complement official ones, helping authorities and humanitarian organizations to better assess when and where to intervene on the ground.

## Introduction

Migration is an ubiquitous phenomenon in human history. People move to improve living conditions or, simply, to escape social distress [1] and natural disasters [2]. The collection of data on migration flows dates back at least to 1871, when the United Kingdom registered the difference of inhabitants in a period of a decade [3]. The data showed that the population changes could not be explained by the number of births and deaths alone, hence another reason had to be involved in the process: migration. Official data on migration flows relies on the comparison of heterogeneous records across countries, which usually have different time scales and time coverage [4]. Data sources include national census, which are released every 10 years, specific surveys, border control and residence permits requests. Surveys and census records are helpful tools for estimating migratory statistics, but the information is not provided at a multinational scale, they are costly and, consequently, the update times tend to be long. Looking at a single country, the net population variation, besides changes due to births and deaths, can be calculated as the difference between the number of immigrants and emigrants. Migrants' mobility, however, can involve several countries biasing the single-country statistics. There can be multiple entrances in a country by the same individual or several countries along

of the Balearic Islands and the European Social Fund with grant code FPI/2090/2018. AT acknowledges financial support from the AEI, Spanish National Research Agency, with grant code PTA2017-13872-I and the Government of the Balearic Islands. MM, AT, PC and JJR also acknowledge funding from the Spanish Ministry of Science, Innovation and Universities, the AEI and FEDER (EU) under the grant PACSS (RTI2018-093732-B-C22) and the Maria de Maeztu program for Units of Excellence in R\&D (MDM-2017-0711). We acknowledge support of the publication fee by the CSIC Open Access Publication Support Initiative through its Unit of Information Resources for Research (URICI).

**Competing interests:** The authors have declared that no competing interests exist.

the trajectory of migrants. This adds inconsistencies to the information at the local perspective, e.g., recurring and returning migrants can affect the number of border crossing events. To capture the complexity of the migratory phenomenon, the concept of interregional flows must be introduced [5]. In this context, another potential data source could be air traffic records that allow to estimate human mobility worldwide [6]. However, they only capture one transportation mode, long or medium distance movements and there exist a bias toward wealthier individuals.

Given this situation, there has been recent calls for new sources providing reliable data on the mobility of migrants and, especially, refugees [7–10]. Data associated to the use of information and communication technologies (ICT) has experienced a large growth in the last decade. ICT data contains temporal and spatial records and it is a useful, fast and inexpensive information source to characterize human mobility at different scales [11]. For example, mobile phone records have been employed to study human mobility with promising results at urban [12–15] and inter-urban level [16–18]. They have been also used to analyze migrant communities distribution and integration in cities [19, 20]. Yet the phone sector is too fragmented to cover a global scale since the data is usually restricted to a single country, making cross-border movements hard to observe. Other examples with similar, yet even more reduced geographical scope, are the GPS tracks left by cars [21].

To expand the focus beyond national borders, we need data coming from services that are genuinely transnational. The widespread adoption of online social networks has finally introduced such global dimension. For instance, the advertising tool of Facebook has been used as a source of migration data although the internal user classification of the platform is not very transparent [22]. In a more open context, data from Twitter has been used to uncover worldwide patterns of human mobility [23–25], infer international migration flows [26–28], estimate cross-border movements [29] and study immigrant integration [30, 31]. Twitter data is also available in countries where census data is unaccessible, outdated or only attainable in the local language [32]. Furthermore, Twitter data has been shown to bear mobility information compatible with the one provided by cell phone records in cities [33]. On the other hand, the data is sparse (i.e., users are not continuously active) and there are biases towards younger and richer individuals [33–36]. This implies that, when taking into account smaller scales, a thorough validation exercise is necessary. Although geolocated Twitter data is sparser than census, surveys and mobile phone records, the observed level of correlation allows for the interchangeability of these sources to study population density and mobility [33, 37, 38].

In this work, we introduce a method to uncover migration flows using Twitter data. As a proof of concept, we focus on the current Venezuelan migration crisis. At country level, the flows obtained from Twitter have been validated against official estimations released by the International Organization for Migration (IOM) in September 2018, the Federal Police (*Policia Federal*) of Brazil and by the United Nations High Commissioner for Refugees (UNHCR or ACNUR) in November 2018 and January 2019. Furthermore, our method provides information at finer geographical and temporal resolutions. It also allows the exploration of other issues not contemplated in official records, such as mobility across multiple countries, routes taken by the migrants, as well as the places where they settle down.

## Materials and methods

### Classical data sources

Traditionally, official statistics come from census surveys, household or labor surveys, population registers, administrative records and border control. For sake of completeness, we briefly summarize the characteristics of these sources and offer references for further information. In

terms of surveys and starting by census, surveys of census focus on migrant stocks or migrant flows including socio-economic features like the country of birth, citizenship, age, sex, education, occupation and time of arrival of migrants in the new country [39, 40]. Spatial information in census surveys is typically given at regional scale but modern censuses provide information even at finer scales. The typical limitations of censuses on migration data are their cost, the slow updating (usually every 5 or 10 years), which decreases the information validity since it gets outdated fast, and the fact that they do not capture illegal immigration. Secondly, household surveys, apart from measuring migrant stocks or migrant flows, focus on the drivers and the impact of migration, internal displacement, emigration and immigration of a given country. A third source comes from labor force surveys which produce statistics on migrant stocks in the labor market of the country. These data allow for country by country comparability and they offer detailed data on small population groups. However, many countries do not include crucial migration questions in their census surveys such as citizenship, country of birth and year of immigration of the person, they may be infrequent or costly and can also present issues with sample size and coverage.

Population registers are held by local authorities and, as the surveys, provide information on the people living in the area. However, there are differences on how the registers are designed, how they define residence, on the population coverage sometimes excluding foreigners, e.g. citizens of some countries are not required to register inside EU states and sometimes leaving migrants are not required to unregister. As a result, they lack of comparability from country to country. Other important and classical sources of migrant flows and stocks gathered in a different way comes from administrative records, which can include records of citizens changing their usual residence, tourist visas, work visas, study permits, residence permits and work permits. Even if these records are continuous and comprehensive, they lack compatible definitions among different countries, moreover they often lack coverage and availability in some countries. Further issues may come from the fact that this data often does not cover naturalized or illegal residents who overstay their visas. Data may not represent the total number of immigrants in the country, e.g., if the visa granted to the head of the family covers his or her dependents and, as a further example, authorities may not track renewals or changes of citizenship. Lastly, border post records of leaving or entering people keep track of flows between countries. These records often do not discern between migration flows and other kinds of mobility, such as tourism. Moreover, in places where the possibility of evading the border control is high, the records become highly unreliable. A review on these classical methods can be found at [41, 42]. In particular, here we employ official data from border control, permits statistics and registers gathered by the UNHCR [43, 44], the IOM [45] and the Federal Police of Brazil [46, 47] to validate the information obtained from Twitter.

## Twitter data for mobility information

Classical data sources have major caveats in terms of comparability, coverage and immediacy of the furnished information [7–10, 48]. ICT data has the potential to complement classical sources by generating estimates of socio-economic phenomena such as transport, mobility, urbanism and migration. Of all the possible ICT data like mobile phone records, social networks, etc., Twitter is one of the most accessible since the data is openly distributed. Twitter data is not safe from biases, several studies demonstrated that people tweeting are mostly young, wealthy and tend to live in cities [34, 49]. However, more recent studies show that the relevance of some of these biases like those related to age or geographical origin are slowly decreasing [35, 49].

Beyond demographics, the impact of biases in the study of aggregated mobility from geolocated tweets has been addressed in [33, 37]. Origin-destination (OD) matrices obtained from Twitter, mobile phone records and mobility surveys were found to be equivalent at scales larger than one square kilometer. This is due to the fact that mobility patterns are not strongly dissimilar across age groups and gender [50] and the demographic biases do not influence so strongly the OD matrices extracted. A similar effect could be expected in refugees and migrant displacements that may be carried out in groups, and thus detecting some members could be enough to describe the mobility patterns.

## Accessing and filtering Twitter data

The data was accessed using the publicly available Twitter Streaming API [51], selecting tweets with geographical information (a description of the script employed is included in Availability of data and materials). We have collected data from January 2015 to December 2018. The location has to be provided at the level of place with a bounding box smaller than 40 $km$ on the longest side, otherwise the tweets are not included in the spatial analysis but we still use them to count country-level flows (using the country identification in the tweet). A minimal filter for bots is implemented by neglecting users tweeting more than twenty times per hour on average in their tweeting life span. Multi-thread tweets are counted as a single one.

The analysis focuses on South and Central America and Caribbean, where over 70% of the Venezuelan migration is concentrated, according to the UN [52]. We divided this extensive territory in a grid of cells of 40 $km$ side, which became our basic spatial units. The position of the users is approximated by the centroid of the cells where they tweet from. With this information, we calculate the radius of gyration of each user trips:

$$r_g = \sqrt{\frac{1}{n}\sum_i^n (r_{cm} - r_i)^2} \qquad (1)$$

where $n$ is the number of tweets, $r_i$ the cell centroid and $r_{cm}$ stands for the center of mass of each user movements. We disregard users with $r_g > 5000$ $km$ (see the distribution of $r_g$ in the Supporting Information) because this implies all the trips are long distance. These accounts are typically multiuser, institutional or company, and cannot offer any mobility information.

## Ethics statement

As a final comment, we adhere to strict data responsibility and ethics principles, ensuring that no personally identifiable information is kept. The meta-data of the tweets, which can be considered personal, are deleted before storage and the user ID is irreversibly encrypted using a SHA-512 algorithm. We do not track individuals, all the spatial analyses are performed using aggregated trips and number of residents at a minimal scale of 1 $km^2$ and in most of the cases over 1, 600 $km^2$. All the figures and tables show aggregated data and only when the number of individuals is larger than three in a cell.

## Resident classification

There is no unique definition of resident applicable to a situation with sparse data as ours. Given the lack of clear definition, we provide several definitions and check if they bear differences in the outcomes of the analyses. Every tweet has an associated country code reference. Users tweeting from Venezuela less then five times in the period 2015-18 or with all the tweets posted in a time window shorter than three months were excluded from the study as non-representative data. After that for each user we identified the most common country of origin of

the tweets posted every month. By doing so, we created the users' history as a sequence of country flags, with one flag for each month. In particular, if the most common country was Venezuela, we labeled the corresponding month as "Ven". If, for a given month, there were no tweets posted or there was a draw between Venezuela and another country, we labeled the month as "Und" (for 'undetermined'). Then we considered the following criteria to discern Twitter Users which are Venezuelan residents (TUVs) based on filters with a gradual level of restriction:

- **The first criterion** is the most restrictive one. We considered TUVs the users appearing to be flagged Venezuela three months in a row at least once in our database. In this way, we obtained approximately 160, 000 TUVs.

- **The second criterion** is slightly less restrictive. It included the users identified before and those with a sub-sequence Ven-Und-Ven. With this method we estimated around 200, 000 TUVs.

- **The third criterion** is similar to the previous one. We included all the users coming from the first criterion and added those with a sequence Und-Ven-Ven or Ven-Ven-Und. This yielded around 216, 000 TUVs.

- **The fourth criterion** is comprehensive of all of the above. With this method, we obtained around 253, 000 TUVs.

Determining the number of TUVs is crucial not only to measure the displacement flows but also to infer the upscaling factors as discussed in the subsection on upscaling factors. These factors allow us to translate the observed TUV numbers into the total flows. We estimated them as the ratio between the population of the country projected from the census and the number of TUVs, updated every year in terms of empirical flows recorded.

## Definition of migrants

The UN definition of long-term migrant requires the person to stay in the new country for at least 12 months. However here we rely on the term of "migrant", which is a term not defined under international law. A migrant is generally defined as a person who leaves the usual place of residence, within the country or crossing a border, in a short or long term and for many reasons [53]. The number of migrants estimated by UN agencies in a single country often relies on classical data, such as records referring to residence permits or flows observed at the border. The objective in this work is to capture flows of people leaving a country in a humanitarian crisis. Definitions based on long-time scales are not adequate for describing population mobility under these circumstances and it does not exist a universal definition of migrant [53]. Therefore, we consider here as migrant any individual leaving Venezuela during the time window of observation, concordantly to the general definition of migrant given by the IOM [53].

## Results

### Upscaling factors

The number of TUVs identified with the four criteria and detected as active (posting tweets) every year is shown in Table 1. As shown in the Table, independently of the criteria used the number of TUVs is relatively stable in 2015 and 2016, and then we found a decline of over 20% in 2017 and almost 50% in 2018. The migration crisis had the first peak in the last months of 2016, so there is a combination of factors that can explain the drop in the number of TUVs. The first one is migration to other countries –estimating this is our target–, which can dissuade

**Table 1. Number of TUVs detected according to the four criteria discussed in the subsection on resident classification in thousands of individuals.**

| Year | *crit*.1 | *crit*.2 | *crit*.3 | *crit*.4 |
|------|---------|---------|---------|---------|
| 2015 | 106 K | 129 K | 134 K | 157.5 K |
| 2016 | 110 K | 132 K | 137 K | 159.5 K |
| 2017 | 82 K | 98 K | 100 K | 116 K |
| 2018 | 51 K | 60 K | 62 K | 71 K |

users from using geolocated social networks, but this factor alone does not explain the entire complexity of the change. We see a drop of 43, 000 TUVs with criterion 4 between 2016 and 2017, while the observed TUVs leaving the country with the same criterion 4 in 2016 is 11, 000 TUVs. To better assess whether this decline of geolocated tweets is a specific behavior of the Venezuelan population or a general trend in the region, we execute the same analysis with Colombian residents as a base line. As a proof of concept, the number of geolocated users residing in Colombia is analyzed using the corresponding criterion 4 for this country. We observe a drop in the same years from 2016 to 2017, from 236, 000 to 198, 000. A general drop in the use of geolocated tweets seems to be happening in the region as a general trend. Factors such as the general use of geolocated Twitter, as well as economic and social stress, must have contributed to such a systematic decline.

The population of Venezuela in 2011, according to the census, was $P(2011) = 29$ million inhabitants [54]. The United Nations Population Division projects the Venezuelan population at around $P(2015) = 30.1$ million for 2015 [55]. As a simplifying assumption, we neglect the population changes due to the difference between natality and mortality. Given the short period of time considered, its effects on the total population are much weaker than those introduced by migration.

The number of active TUVs that same year according to criterion 4 was $u_4(2015) = 157$, 500. This gives us an upscaling factor of $S_4(2015) = P(2015)/u_4(2015)$. For the following year, we have a number $e_4(2015)$ TUVs leaving the country according to the same criterion 4. This implies a population decrease given by $P(2016) = P(2015) - S_4(2015)\, e_4(2015)$. In general, we can write a recurrent formula for the upscaling factor associated to criterion $k$ ($k = 1, \ldots, 4$):

$$S_k(t) = \frac{P(t-1) - e_k(t-1)\, S_k(t-1)}{u_k(t)}, \tag{2}$$

where the first year corresponds to 2015 with $P(2015) = 30.1$, as stated before. Applying these calculations for a given criterion $k$, we obtain an upscaling factor per year. These factors are the inverse of the population fraction tweeting with geolocation during the corresponding year and, with all the needed cautions on possible biases, one can assume that for criterion $k$ one TUV leaving the country represents $S_k(t)$ actual migrants. Measuring the TUVs detected abroad in 2018 and upscaling the flows according to the first exit year, we obtain the numbers shown in Table 2 and in Fig 1.

## Validation of external flows

We are interested in estimating the number of Venezuelan migrants in each country because these are the numbers reflected in the statistics of entries and registers, and constitute the basis for the records used by national and international authorities. Note that the official sources are highly heterogeneous in nature and vary from country to country. The method to define a previously classified Venezuelan resident as a migrant is to detect at least one tweet from them

**Table 2. Estimated migration of Venezuela citizens in the four neighboring countries from which a humanitarian crisis has been reported.** The different lines in "Data" correspond to the different criteria for establishing Venezuelan residents with the Twitter data and the last four lines correspond to official figures from IOM, the United Nations (UNHCR) and the Federal Police of Brazil (PFB). The units are thousands of individuals (K).

| 2018 | Brazil | Peru | Ecuador | Colombia | Total |
|---|---|---|---|---|---|
| Data(1st) | 168K | 589K | 247K | 1,080K | 3,970K |
| Data(2nd) | 163K | 588K | 242K | 1,070K | 3,920K |
| Data(3rd) | 163K | 593K | 246K | 1,090K | 3,960K |
| Data(4th) | 160K | 590K | 243K | 1,080K | 3,920K |
| IOM Sep18 | 75K | 414K | 209K | 935K | 2,600K |
| UNHCR Nov18 | 85K | 500K | 220K | 1,000K | 3,000K |
| UNHCR Jan19 | 96K | 506K | 221K | 1,100K | 3,400K |
| PFB Dec18 | 199K | – | – | – | – |

abroad as a proof of border crossing. This means to count for each year the number of TUVs appearing for the first time in a different country and determine the migrant flows by upscaling as discussed before. Some of the TUVs can be passing through and continue the travel to third countries, where in turn, they will be counted as well. The comparison of the flows obtained in Table 2 with the numbers provided by the international and national agencies shows a good alignment with our estimation. An impression further confirmed in Fig 1 for the UNHCR data in most of the countries in the area with $R^2$ over 0.9. The main outlier in the Table 2 is Brazil, where the IOM and UNHCR give values well below our estimations. However, the Federal Police of Brazil registered a total number of entries over 199, 000 from January 2017 until December 2018 (even though half of them end up returning to Venezuela [46]). Our numbers lay within the range provided by the different sources of information. Given the

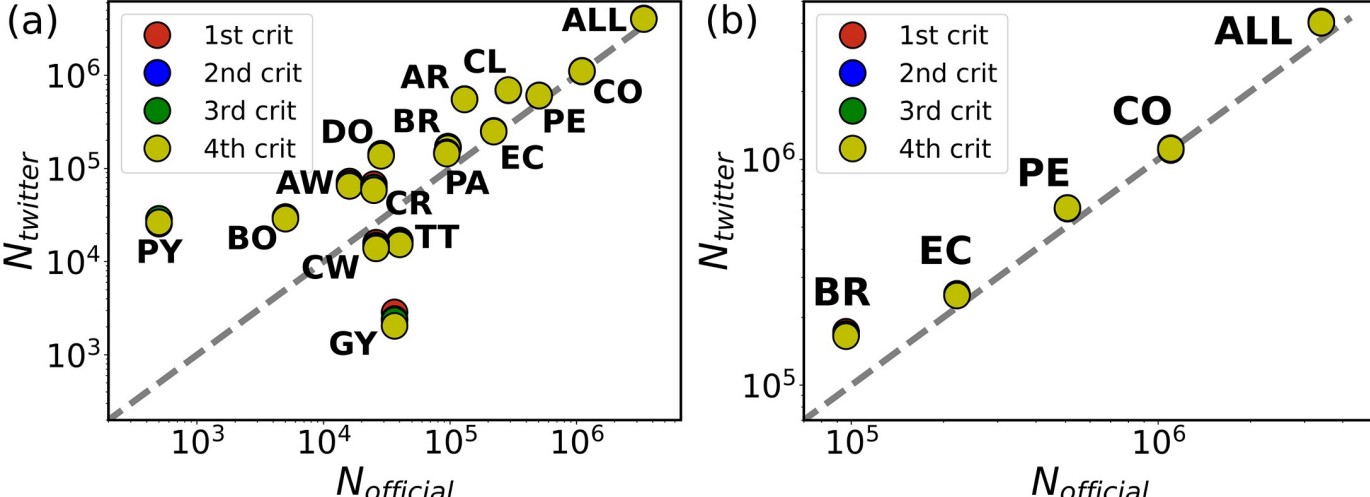

**Fig 1. Country-level flow validation.** Comparison between the migrant flows estimated from the upscaled Twitter data $N_{twitter}$ following the different resident criteria and the official numbers $N_{official}$ from the UNHCR in January 2018. These official numbers in some countries are projections. Each point corresponds to a country. The panel (a) shows all the countries of the considered area, and the panel (b) specific detail for neighboring countries from which a humanitarian crisis has been reported. In both cases, the grey dashed lines are the diagonal. The correlation produces a $R^2 = 0.98$ for all the countries and $R^2 = 0.99$ for Brazil, Colombia, Ecuador and Peru alone. The country codes are Argentina (AR), Aruba (AW), Bolivia (BO), Brazil (BR), Chile (CL), Colombia (CO), Costa Rica (CR), Curaçao (CW), Dominican Republic (DO), Ecuador (EC), Guyana (GY), Panama (PA), Paraguay (PY), Peru (PE) and Trinidad and Tobago (TT). The panels are displayed in log-log scale due to the several orders of magnitude of the flows. However, the important point here is to verify the identity between expected and measured values and, therefore, the correlation analysis is performed with $R^2$ in the original scale.

similarity across the upscaled flows obtained with the different criteria, we continued our analysis with criterion 4, which provides the largest statistics.

## Migration routes

Beyond flow estimations, it is important to identify the preferred routes taken by migrants in order to better provide targeted humanitarian assistance. As mentioned, we took as geographical basis a grid of cells of 40 km side covering the full South American continent, the Caribbean and part of Center America. We counted the cumulative number of TUVs posting messages in each cell. The number of distinct TUVs in the whole period (2015-2018) is plotted as a heat map in Fig 2. The densest areas are located in Venezuela, specifically near Caracas. The relevant users for the analysis are those classified as residents (TUV) and who have been detected abroad. The beginning of most of their trips are in Venezuela and the density matches population distribution. Once in other countries, the routes concentrate also in the large cities like Buenos Aires, Santiago, São Paulo, Rio de Janeiro and Lima (which can be temporary stops or travel destinations) and around the principal roads and rivers. For example, following a preferred route from Venezuela to Manaus, a split occurs: some TUVs take the ferry navigating the Amazon to the Brazilian city of Belem and subsequently travel along the coast to the cities of Fortaleza and Salvador; while other TUVs travel from Manaus through the forest via the BR-319 highway towards the border with Peru and Bolivia. Overall, the preferred migration route in South America is the Pan-American highway, passing through Colombia, Ecuador, Peru and all the way to Chile and Argentina. Other minor routes are also detected, such as the one from La Paz in Bolivia to Cordoba in Argentina.

To better understand the potential application of our method, we built a vector representation in each cell. For every TUV tweeting from the cell, we built a unit vector pointing from the present location to the cell with the consecutive tweet with the condition that it must be closer than 500 km to exclude as much as possible air traveling and noise coming from infrequent users. We separated the unit vectors in those pointing towards or away from Venezuela, and we added those in every cell in each category. The resulting outgoing vectors are displayed in red in Fig 3, while the in-going ones are in blue. These maps provide information on the main ground exiting routes from Venezuela reported from official agencies and also on the total flow observed in both directions. To go further, we can establish a line intersecting the routes (center of the parallel dashed lines in Fig 3) and consider the upscaled number of TUVs to calculate the number of people crossing the line per year and the direction they are going. When calculating these numbers, we are not using the vector representation. To be sure that these users effectively crossed the given line, we impose as a necessary condition to have them tweeting on both sides of the dashed line within circles of 350 km radius. For example, in Fig 3b, the dashed line is placed halfway between Boa Vista and Manaus, the first circle comprehends an area of the radius of 350 km around Boa Vista, and the other one spans the same area around Manaus. In Fig 3a, the same is done between Bogotá and Quito. The results for 2017 are shown in the right-bottom corner of the maps, while the information collected year by year is included in Table 3.

It is important to note that the numbers are relatively small. This is due to the fact that we need to see two consecutive tweets: one above and another below the line to identify a user. However, not all TUVs have these two tweets, as some may travel directly from Venezuela by plane or simply do not tweet so often during their trips. These numbers are, therefore, underestimations although they are proportional to the total flow. We would need to further upscale them according to the fraction of TUVs active enough to be detected on both sides of the lines. Without further processing, we can, nevertheless, compare results obtained on the same route

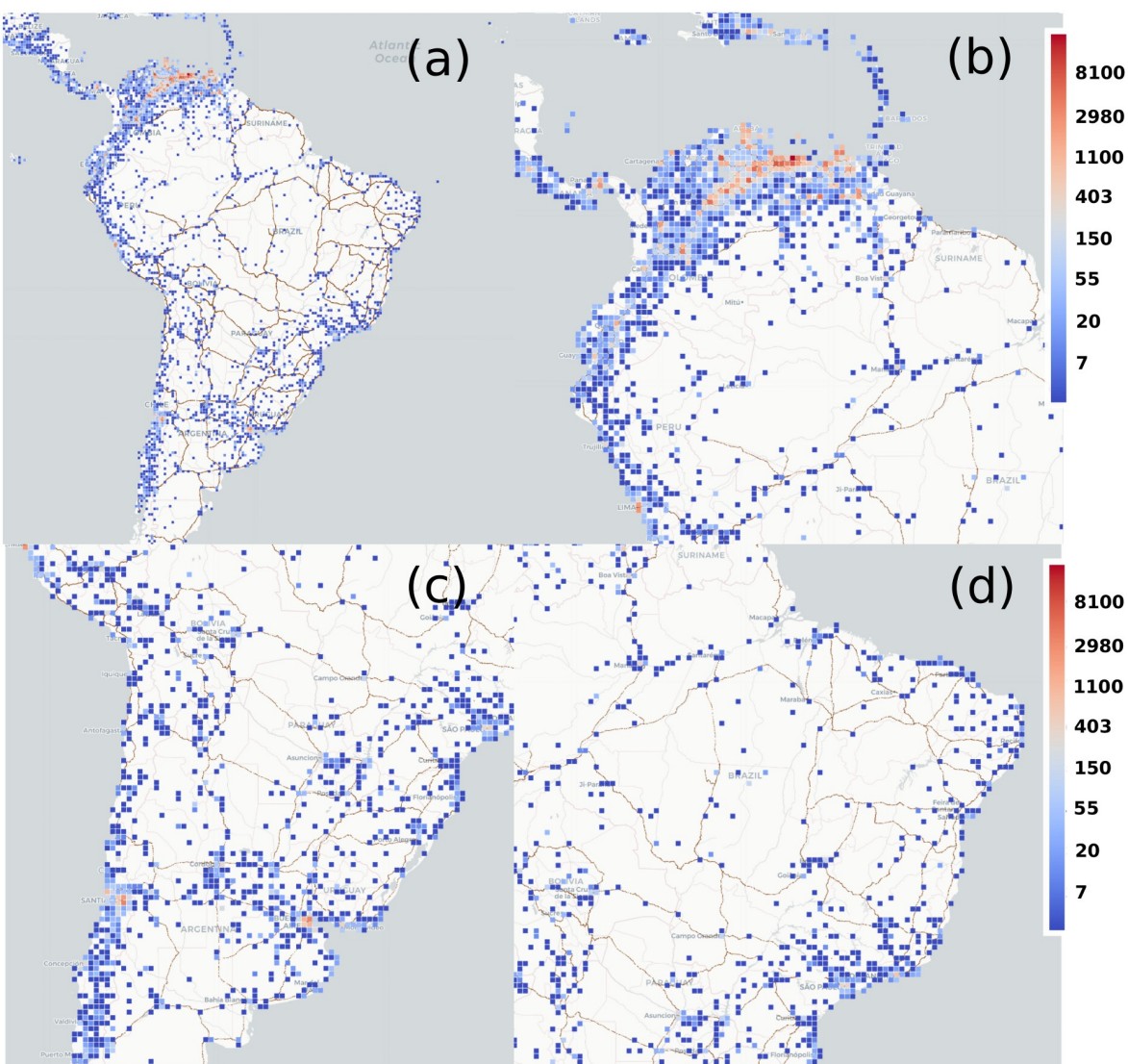

**Fig 2. Migrants' routes.** Number of individuals observed in every $40 \times 40$ km$^2$ cell in the area of study. The heatmap scale is logarithmic. Only cells with more than 3 individuals are displayed. In (a) the full South American continent plus the Caribbean and Center America; (b) A zoom in on the Northern area focused on the Caribbean; In (c), a zoom in highlighting the Southern Cone; And in (d), a zoom in on Brazil. Map tiles by Carto, under CC-BY 4.0. Data by OpenStreetMap, underODbL.

year by year and between routes with the same methodology. In Table 3, we observe a clear domain of the outflows over the inflows to Venezuela. In the Pan-American highway, there is an important grow in the flows year by year, almost doubling in the last period between 2017 and 2018. The route entering in Roraima has, in turn, a flow that is close to a factor 10 below the Pan-American one, with an initial increase until 2017 followed by a strong decline in the following year. The latter case may be explained by the fact that approximately half of the migrants entering Brazil have been registered to leave in the next months [46, 47].

## Recurrence

Note that 25% of the migrant TUVs travel back and forth from Venezuela (recurrent travelers) as we will show below, while the remaining 75% stay most of the time abroad. To estimate the

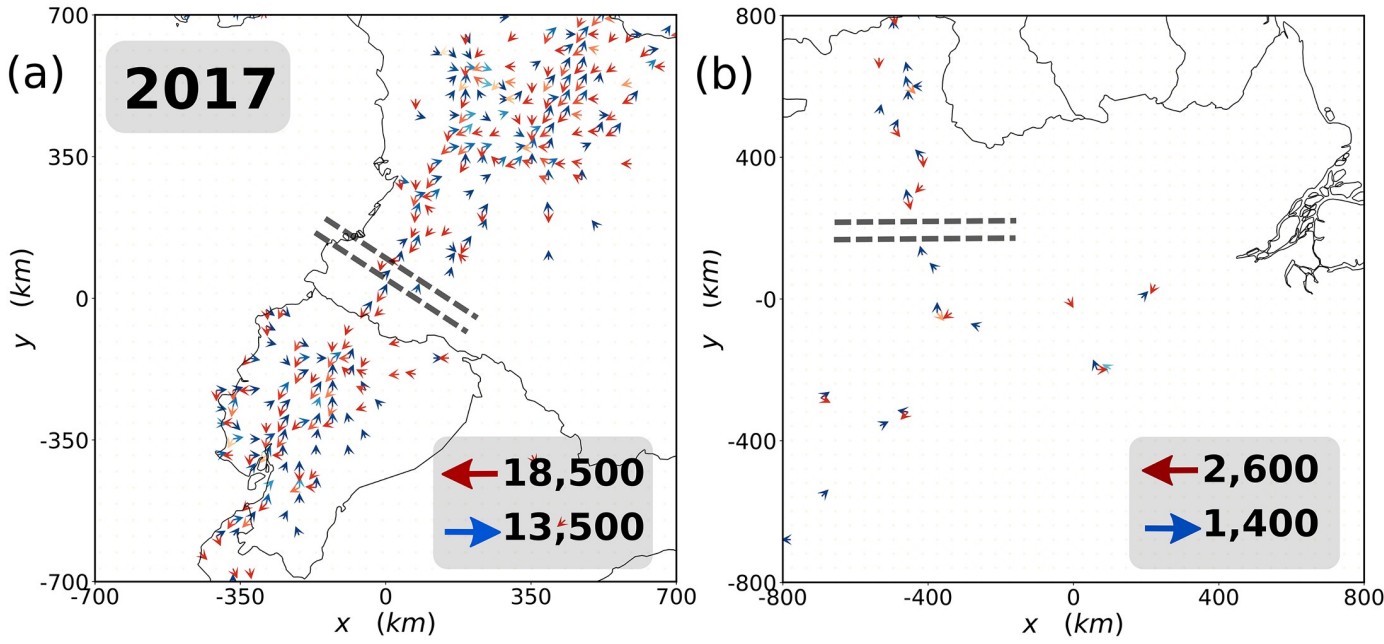

**Fig 3. Crossing routes.** Map of two main ground migrant exit routes from Venezuela reported by official agencies: in (a) the Pan-American road with a portion of Colombia, Ecuador and Peru, and in (b) the area of Roraima, Amazonas and Pará states in Brazil. Blue arrows indicate flows toward Venezuela and the red ones away from it. Only cells with more than three TUVs are shown in the maps. The lightness of the color of the arrows is proportional to the net in- and out-flows from light to darker colors. The upscaled net flows crossing the dashed lines are displayed in the right-bottom corner of each plot.

time spent in a location, we take the following convention: the time between consecutive tweets is assigned to the location of the first one. Applied to countries, this allows us to define the cumulative time spent abroad for each TUV after his/her first travel to other country, $t_{out}$. Additionally, we can calculate the time span between the first tweet abroad and the last tweet of the user, $t_{Tot}$. In this way, we define a ratio between the time spent abroad and the total after the first exit from Venezuela for each TUV, $R = t_{out}/t_{Tot}$. There is a wide range of behaviors as can be seen in the distribution of Fig 4. TUVs detected abroad for the first time in their last tweet are not considered. Note that the precision is limited by the heterogeneity in the user inter-event time distribution and the total time window that we are able to analyze. Some TUVs correspond the canonical view of migrants, leaving the country and coming back only seldom, while others go back and forth. More recent migrant TUVs may be classified as not recurrent, even though they might be correctly classified if observed for longer time periods. Even so, we need to establish a criterion to discern between frequent returners and those staying mostly abroad. The distribution of $R$ shows two clear peaks at the extreme values of the domain with a valley in the region between 0.4 and 0.75, so the threshold is set at 0.5. Results are similar for other values of the threshold provided that they are in the range [0.4, 0.75]. We find 12, 518 TUVs classified as recurrent and 22, 459 as non recurrent.

**Table 3. Upscaled number of TUVs crossing the lines of Fig 3 away (←) and toward (→) Venezuela in the two routes considered.**

|  | Year | 2015 | 2016 | 2017 | 2018 |
|---|---|---|---|---|---|
| Pan-American road | away (←) | 6,550 | 11,300 | 18,500 | 36,200 |
|  | toward (→) | 5,500 | 9,300 | 13,500 | 23,700 |
| Roraima | away (←) | 1,200 | 1,490 | 2,600 | 1,070 |
|  | toward (→) | 1,200 | 1,300 | 1,400 | 720 |

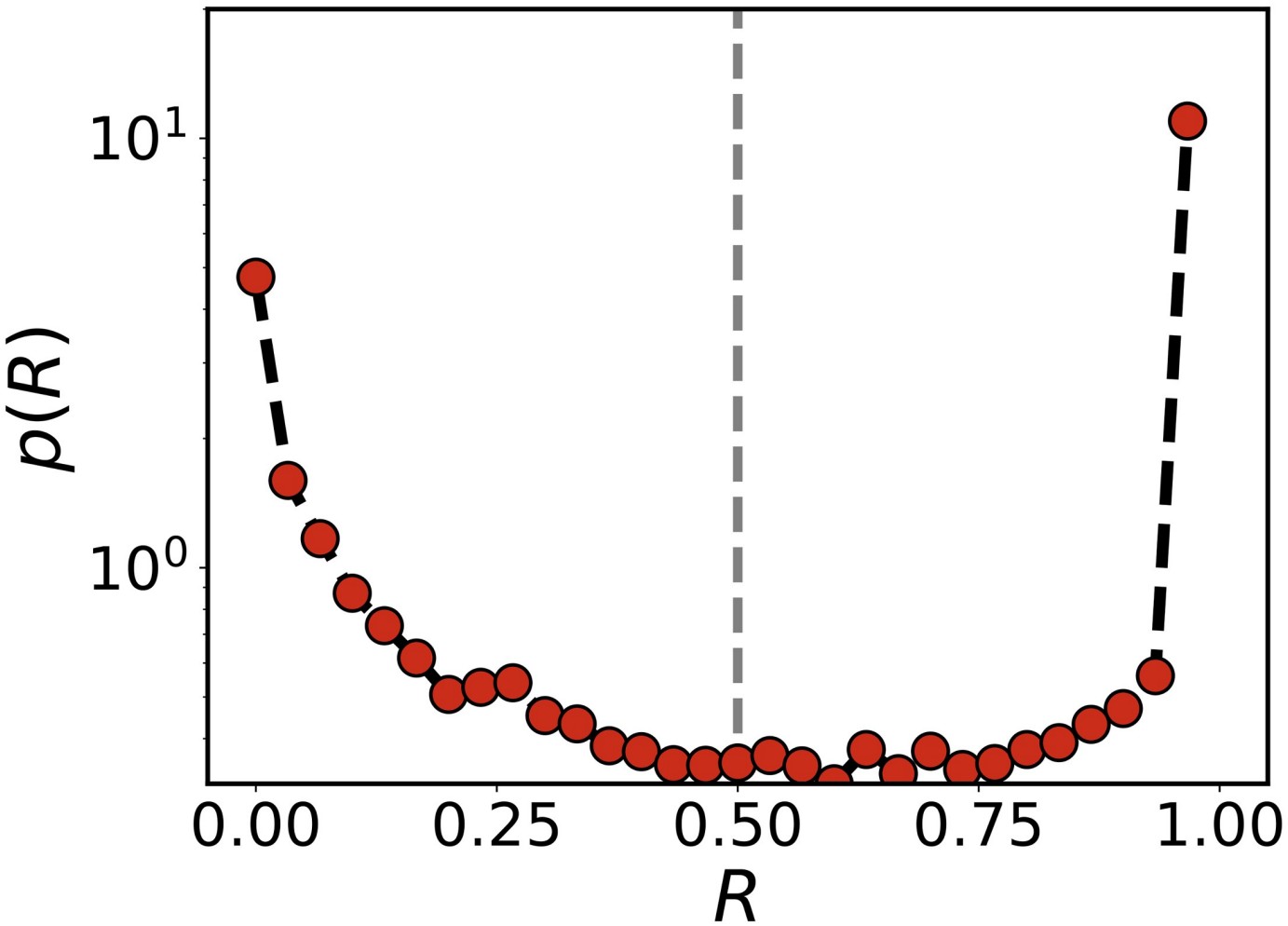

**Fig 4. Time spent abroad.** Probability distribution of the fraction of time spent abroad after the first country exit. TUVs with *R* lower than 0.5 are classified as recurrent.

Recurrent TUVs stay most of the time in Venezuela, so it can be assumed that they still reside somewhere in the country. Similarly, non recurrent individuals are likely to have a residence place abroad. We define as residence place/country the location from which they tweet most in the last month of activity. Out of the 22, 459 non recurrent TUVs, 16, 292 have a place of residence assigned in the geographical area of our analysis out of Venezuela. To upscale these numbers in each of the countries, we apply the same technique as in previous sections, see Eq (2). The factor is calculated as the ratio between the updated Venezuelan population and the number of TUVs in each year. The number of migrant TUVs in every country is upscaled according to the factor corresponding to their first year of exit. The results per country are shown in Table 4 and compared with the official estimations from international agencies. The numbers of Table 2 refer to entries across the border, a single individual can contribute to more than one country. In contrast, the records in Table 4 assign one country to each migrant and the numbers contain only distinct individuals in the full study area. Our method provides numbers below the official statistics in Brazil, Ecuador and Colombia, while in Peru it is slightly higher. We see that the new settlement places concentrate in Argentina,

**Table 4. Estimated number of venezuelan residents in each of the countries in late 2018.** The units are thousands of individuals (K).

| 2018 | Brazil | Peru | Ecuador | Colombia |
|---|---|---|---|---|
| Data(4th) | 58K | 504K | 170K | 826K |
| IOM Sep18 | 75K | 414K | 209K | 935K |
| UNHCR Nov18 | 85K | 500K | 220K | 1,000K |
| UNHCR Jan19 | 96K | 506K | 221K | 1,100K |

Chile, Colombia and Peru, while the flows get through other countries such as Brazil but the fraction of migrant fixation is lower. We have performed as well a systematic comparison between the number of Venezuelan residents in each country provided by our method and by that of international agencies. The result is displayed in Fig 5, where one can see an acceptable agreement with a $R^2$ over 0.97.

## Spatial integration of migrants

In addition to large-scale numbers, the data also allows for a local study on the place of residence of the migrant population. As example, we look at three of the main cities in South America: Bogotá in Colombia, São Paulo in Brazil and Lima in Peru, where the statistics are more reliable. The urban space is divided in a grid of cells with $1 \times 1$ km$^2$ area where we assign to migrants the most common place from which they tweet during night hours (between 8PM and 8AM local time). The resulting heatmaps are displayed in Fig 6. Below, we also show the distribution of the local population obtained from local twitter users to have a comparison basis. Besides a visual inspection, we have calculated the segregation index proposed in [31] called $h$. This metric is the ratio between the entropy of the spatial distribution of the migrant community and that for the local population with a correction to take into account finite size effects. If $h = 1$, both populations are similarly distributed, while smaller $h$ indicates segregation. As a complement, we also calculate the normalized mutual information NMI between the distribution of migrants and locals. The NMI is a way to compare the distribution of two variables and it ranges between 0 (the variables are independent) to 1 (they come from the same distribution) [57]. When applied to the former Venezuelan residents, the results are shown in Table 5. All the values of $h$ and NMI are low. In these cities, Venezuelan migrants are far from being well integrated from a spatial point of view. There are many causes behind this behavior, ranging from housing prices and availability to the presence of migrant communities from the same country. Moreover, the distribution of locals says nothing on migrants' residence places. Hence it is not to be considered a proxy for migrant distribution in the three cities. Specifically, in the case of São Paulo, both metrics are very low, although this must be taken with certain caution because only 50 users were detected there against 1, 300 in Bogotá and 570 in Lima.

## Temporal distribution of upscaled outflows

Considering only outflow from Venezuela, we can unfold a time series of the upscaled number of exits per month. The results are shown in Fig 7a. The monthly numbers start to increase in 2015, peaked in late 2016, and increases later until the end of our time window in January 2019, consistently with the data recorded from the Federal Police of Brazil [47]. The histogram shows some peaks and valleys that can be correlated with special events during the crisis. This can be seen in the lower panel Fig 7b, where we consider the first exit from Venezuela per TUV. In this version, the impact of the events is more clearly appreciated since they correlate better with outflow. In all cases, we are only showing the upscaled outflows. As discussed for the recurrent TUVs, there exists an inflow that partially compensates the exits.

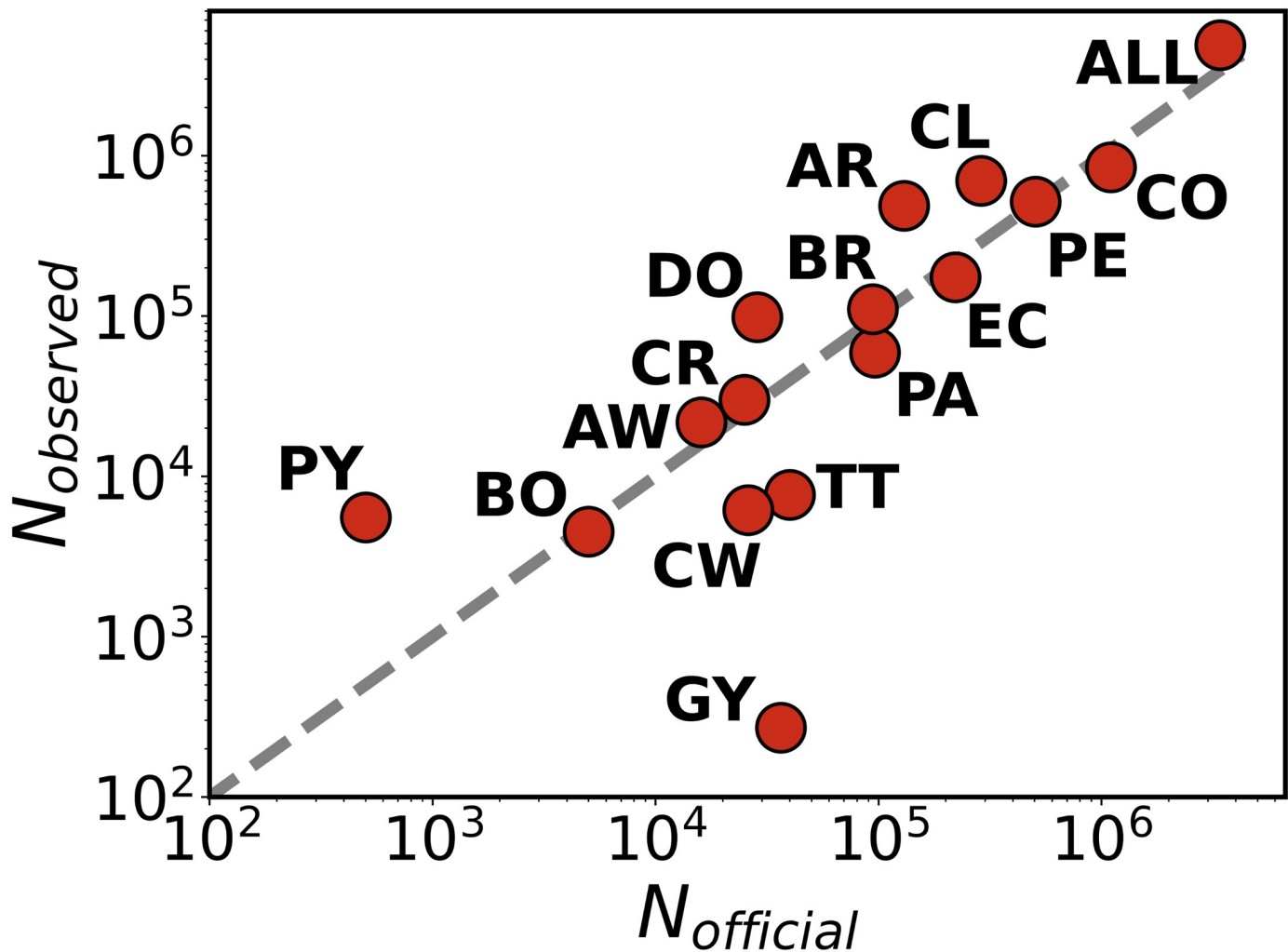

**Fig 5. Validation of new residents.** Scattered plot with the comparison between the estimations of new Venezuelan residents obtained with our method and the official data from the international agencies in each country. Every circle is a country, the dashed gray line is the diagonal. In this case, the correlation is $R^2 = 0.97$. The plot is displayed in log-log scale due to the several orders of magnitude of the number of residents. As before, the important point is to verify the identity and, therefore, the correlation analysis is performed in the linear scale.

## Discussion

According to five UN agencies, massive gaps in data covering refugees, asylum seekers, migrants and internally displaced populations threaten the lives and wellbeing of millions of children on the move. Through a joint Call to Action [58], the agencies confirm the critical need to improve the availability, reliability, timeliness and accessibility of data and evidence to better understand how migration and forcible displacement affect the wellbeing of people. In this work, we have developed a method to contribute to filling these glaring data and knowledge gaps by extracting migrant mobility flows from geolocated Twitter data. We focus on the current crisis in Venezuela, although the method is universal and can be translated to other contexts conditioned only to the data coverage.

The analysis performed here is subject to following considerations, restrictions and assumptions:

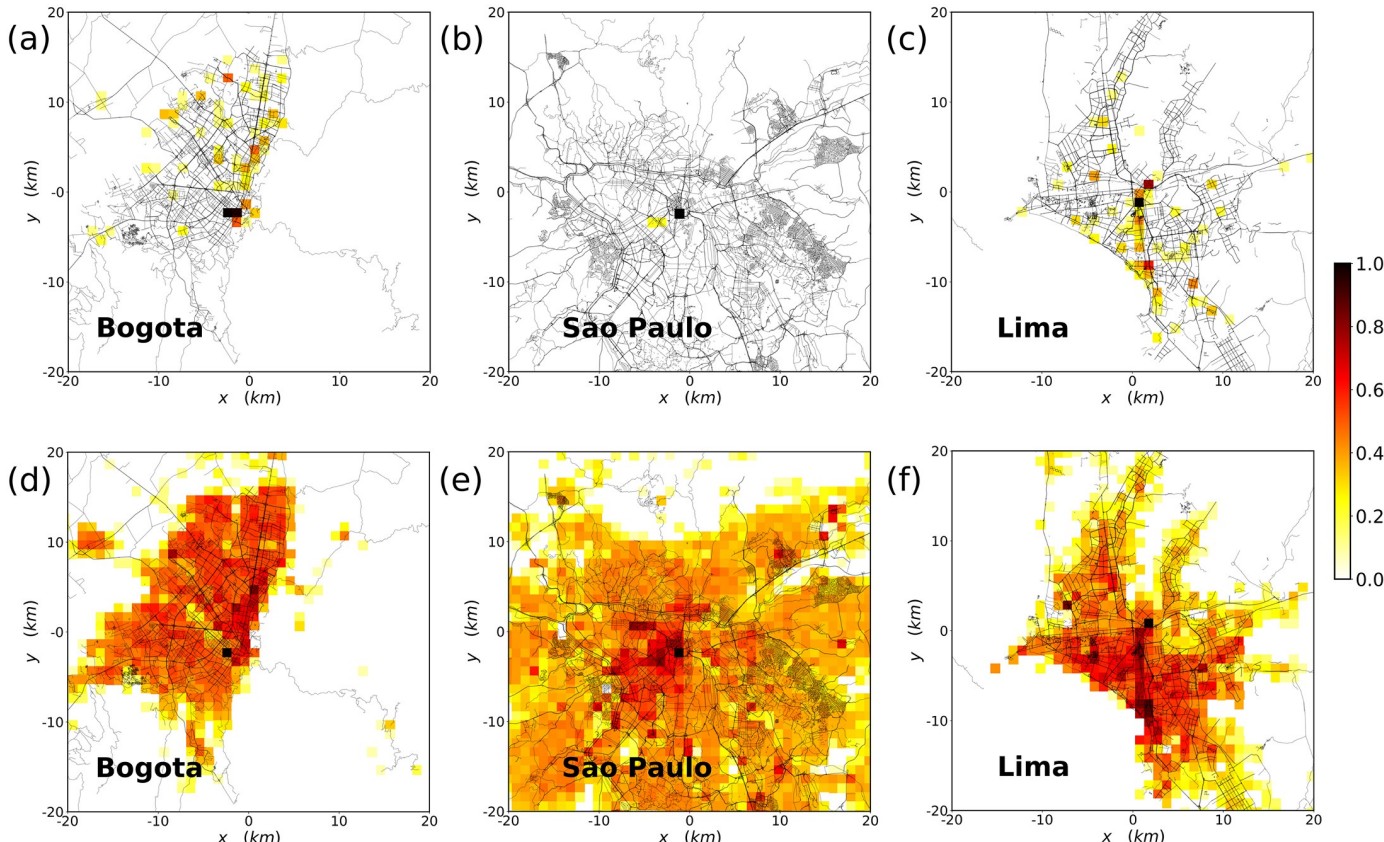

**Fig 6. Residence locations in the main cities.** Log-scale heat map of the population distribution in the main cities of the area. In the top row, the data corresponds to numbers of migrant TUVs and in the bottom to the local geolocated Twitter users. The scale of the heatmap is logarithmic and the maximum is rescaled in each of the maps. In (a) and (d), results for Bogotá (Colombia). In (b) and (e) for São Paulo (Brazil). And, in (c) and (f), for Lima (Peru). Data on roads by OpenStreetMap contributors and from MapCruzin, all available under the Open Database License ODbL. For more information check [56].

- A few previous studies anticipated the potential of Twitter for the study of migration flows [26–28]. The main focus of these works is on the presence of migrants and the estimation of country level migration flows. Here we show the full potential of Twitter geolocated data to capture mobility patterns on the route, finding the specific destinations of migrants within a country and to study the time series of migration flows. We have provided as well a systematic check of different methodologies to define residents and a sound comparison between flows detected from Twitter and numbers offered by international and national agencies.

- Geolocated information in Twitter. There are two types of tweets according to the field containing the metadata on geolocation: tweets with coordinates and those with place. The ones with coordinates are precise within GPS resolution, they are a minority and Twitter is discontinuing their usage due to privacy concerns, even though user's informed consent is required to post them. The tweets with place are a majority among those geolocated [59].

**Table 5. Segregation indicators h and NMI for migrant TUVs.**

|       | Bogotá | Lima | São Paulo |
|-------|--------|------|-----------|
| h     | 0.59   | 0.62 | 0.27      |
| NMI   | 0.05   | 0.07 | 0.04      |

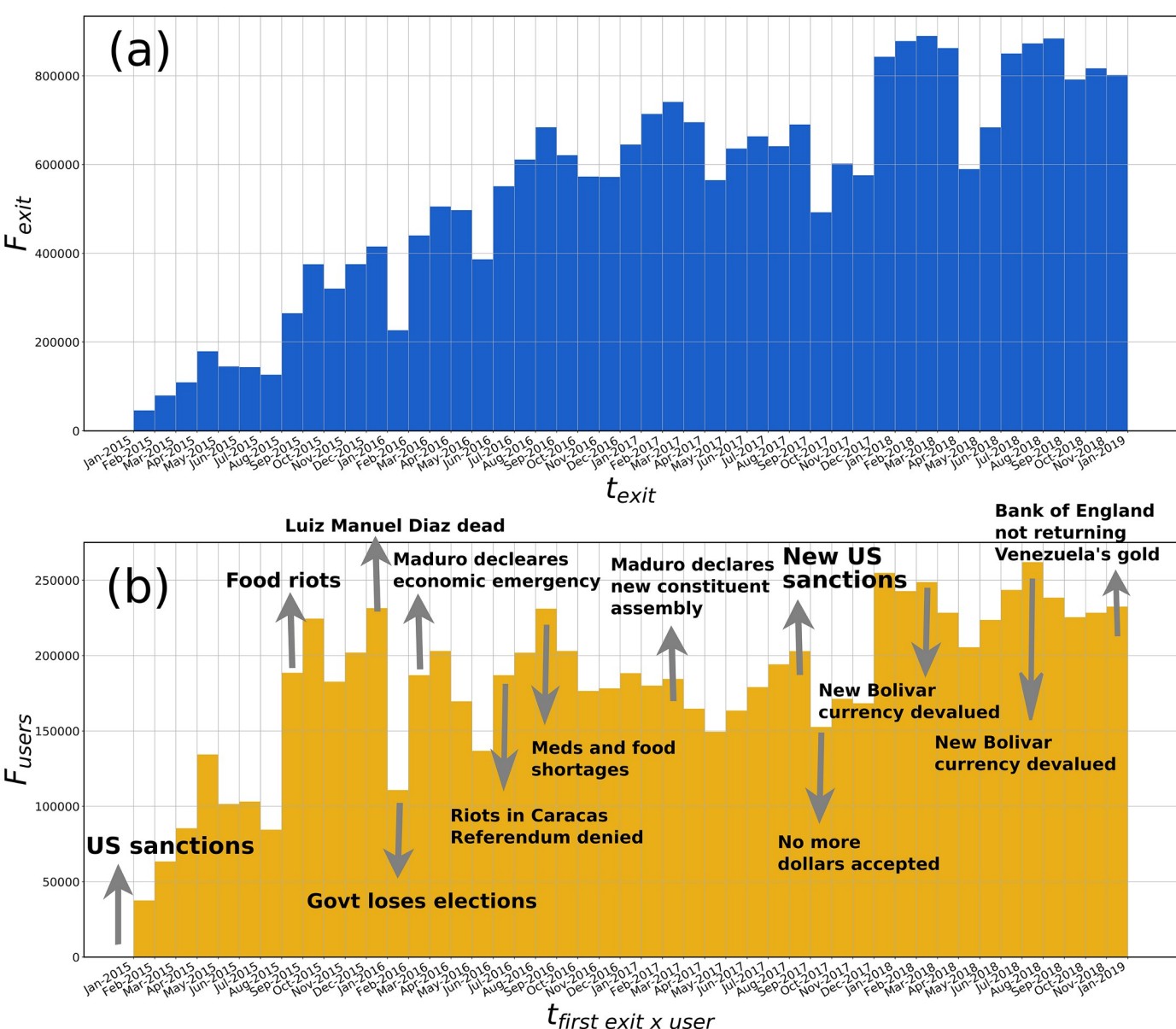

**Fig 7. Exit times distribution.** (a) Total upscaled exits from Venezuela $F_{exit}$. (b) First exits from Venezuela per TUV upscaled to obtain $F_{users}$.

The place refers to a geographical bounding box around the posting position, which may corresponds to neighborhoods, municipalities, provinces or states, etc. Since we want to study migration out of a country, most of these place levels are accurate enough for our purposes and they are used in the analysis.

- Note that in this work the definition of Venezuelan residents comes from temporal tracking of users' location at country level. We have not used methods such as the content of the tweets [60], where the authors manually processed a sample of the tweet texts in order to classify users as refugees or not. The analysis was based on the presence of keywords such as "refugee", "asylum", "camp", etc. Only 5.4% of the users in their text filtered dataset is classified as a refugee, another 16.2% are potential refugees, while the rest are journalists, tourists

and others. Our work is free of these biases because we use a completely different approach based on geolocation only. The analysis of the message contents may be of interest as a byproduct of our work with aim of detecting the main concerns of the migrant population.

- Here we employ the term "migrant" concordantly to the definition given by the IOM [53], which requires the person to leave the original place of residence for a variety of reasons. The method to define a previously classified Venezuelan resident as a migrant is to detect at least one tweet from them abroad as a proof of border crossing, similarly to a previous study [60]. We could have used a stricter criterion and request two or more tweets abroad to classify migrants (and on the other hand request two tweets per year in order to classify residents as active residents). This would not affect the average flows (as we saw from other methods, the upscaling factors absorb and rescale the estimated flows), although it notably narrows the statistics. Hence we discarded this option. On the other hand, our definition relies on the classification of users as Venezuelan residents. We do not have a unequivocal definition and, therefore, we have studied four different methods based on temporal and quantitative sampling, which are shown to be almost identical after rescaling. This can be observed in the way they give similar results when measuring migration flows with respect to classical data estimations from the UN in Fig 1.

- Twitter data provides information from a fraction of the total population and, consequently, it may suffer from biases. Previous studies have shown that these biases in terms of age and socio-economic level have a limited impact on the estimation of mobility flows [33, 37]. These studies were conducted in Europe, where the economic inequalities between rich and poor are less pronounced than in Latin America. Wealthy migrants from Venezuela are able to fly and their destination can include, besides, the South Cone countries in the North of the Americas and Europe. The fact that we detect non-negligible flows of terrestrial movements show that we are capturing also the least wealthy sections of the population. Additionally, we perform a systematic comparison between flows estimated with our methods and the ones offered by official records. The close agreement proves that the analysis is not ignoring significant parts of the population.

- Among the countries with the largest flows of Fig 1, we see an overestimation of migrants in countries of the Southern Cone, namely, Chile and Argentina. Up to our understanding, this distortion may be due to two relevant factors. On one side, the penetration rate of people using Twitter from these countries may be different from Venezuela. From a social perspective, Venezuelans moving to these countries may conform to the local culture regarding a different usage of social platforms. This results in a over-weighted estimation of those migrants who end up tweeting from Chile and Argentina. This may be a factor, but it is unlikely that people adapts so fast to local usages. On the other hand and most importantly, there could be a sub-representation of Venezuelan migrants in the official statistics of these countries. For example, this happens in the numbers of Table 2 with respect to immigrants entering in Brazil. The official UN-agencies estimated almost one half of the inflows registered by the Federal Police. The numbers estimated with our method were much closer to those of the Police.

- Twitter data has the advantage of being publicly available. Still, before using them in operational context, a through validation exercise like the one we have performed here is necessary. Part of the validation could be done against other private data sources such as mobile phone records, but these data sets are usually constrained to a single country and users can change their phone number/provider at border crossing while the online social network accounts remain unaltered.

This work has proved useful for humanitarian agencies, such as UNICEF, in better understanding the magnitude of the Venezuelan crisis in a country of continental proportions, such as Brazil, shaping the design of broader interventions beyond the border-crossing in the North of the country. In addition, the method simplicity, coupled with the open availability and pervasiveness of the Twitter data, can bring a new generation of studies to explore migratory crises from a multidisciplinary point of view. Authorities and humanitarian organizations can thus count with this extra data source to implement more informed protocols of response in humanitarian contexts.

## Availability of data and materials

In this work, we use several data sources: Geolocated Twitter, UNHCR, IOM and Federal Police of Brazil. The access links are included as references in the manuscript. In particular, the Twitter API in Ref. [51], the UNHCR in Refs. [43, 44, 52], the UN population division DESA [55], the IOM at [45], the Federal Police of Brazil at [46, 47] and the Venezuelan census [54].

For Twitter, the geolocated data is downloaded using the streaming API. Every user can access this information by following the instructions of the updated user guide on the Twitter streaming API provided at https://developer.twitter.com/en/docs. We include next a working example of the python code needed to do a query to the Twitter streaming API in a limited geographical BOX enclosed between latitude y0 and y1, and longitude x0 and x1. The aim of this code is only illustrative, the commands can be changed in any moment by the Twitter developers.

```
from tweepy import Stream, OAuthHandler
from tweepy.streaming import StreamListener
CONSUMER_KEY = ''
CONSUMER_SECRET = ''
ACCESS_KEY = ''
ACCESS_SECRET = ''
BOX = [x0, y0, x1, y1]
class MyStreamListener(StreamListener):
  def on_status(self, status):
    print(status)
if __name__ == 'main':
  auth = OAuthHandler(CONSUMER_KEY, CONSUMER_SECRET)
  auth.set_access_token(ACCESS_KEY, ACCESS_SECRET)
  listen = MyStreamListener()
  stream = Stream(auth, listen, gzip=True)
  stream.filter(locations=BOX)
```

## Acknowledgments

We thank Riccardo Gallotti and Daniela Paolotti for useful comments and suggestions.

## Author Contributions

**Conceptualization:** Mattia Mazzoli, Boris Diechtiareff, Antònia Tugores, Willian Wives, Natalia Adler, Pere Colet, José J. Ramasco.

**Data curation:** Mattia Mazzoli, Antònia Tugores.

**Formal analysis:** Mattia Mazzoli, Natalia Adler, Pere Colet, José J. Ramasco.

**Funding acquisition:** Pere Colet, José J. Ramasco.

**Investigation:** Mattia Mazzoli, Pere Colet, José J. Ramasco.

**Methodology:** Mattia Mazzoli, Antònia Tugores, Natalia Adler, Pere Colet, José J. Ramasco.

**Project administration:** José J. Ramasco.

**Resources:** Boris Diechtiareff, Antònia Tugores, Willian Wives, Natalia Adler, José J. Ramasco.

**Supervision:** Pere Colet, José J. Ramasco.

**Validation:** Mattia Mazzoli, Boris Diechtiareff, Willian Wives, Natalia Adler, Pere Colet, José J. Ramasco.

**Visualization:** Mattia Mazzoli, Antònia Tugores.

**Writing – original draft:** Mattia Mazzoli, Natalia Adler, Pere Colet, José J. Ramasco.

**Writing – review & editing:** Mattia Mazzoli, Boris Diechtiareff, Antònia Tugores, Willian Wives, Natalia Adler, Pere Colet, José J. Ramasco.

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
