## [Decision Letter · Decision Letter 0]

11 Dec 2019

PONE-D-19-28022

Migrant mobility flows characterized with digital data

PLOS ONE

Dear Mr. MAZZOLI,

Thank you for submitting your manuscript to PLOS ONE. After careful consideration, we feel that it has merit but does not fully meet PLOS ONE’s publication criteria as it currently stands. Therefore, we invite you to submit a revised version of the manuscript that addresses the points raised during the review process.

Both reviewers consider that your manuscript has merits to be considered for publication. However, they also observe some minor issues that should be addressed. Please respond to all reviewers' comments.

We would appreciate receiving your revised manuscript by Jan 25 2020 11:59PM. To enhance the reproducibility of your results, we recommend that if applicable you deposit your laboratory protocols in protocols.io, where a protocol can be assigned its own identifier (DOI) such that it can be cited independently in the future. For instructions see: http://journals.plos.org/plosone/s/submission-guidelines#loc-laboratory-protocols

We look forward to receiving your revised manuscript.

Kind regards,

Jordi Paniagua

Academic Editor

PLOS ONE

Journal Requirements:

1.

2.  Please remove your figures from within your manuscript file, leaving only the individual TIFF/EPS image files, uploaded separately.  These will be automatically included in the reviewers’ PDF.

3.

We suggest you thoroughly copyedit your manuscript for language usage, spelling, and grammar. If you do not know anyone who can help you do this, you may wish to consider employing a professional scientific editing service.  

4. Please clarify whether there was any ethical oversight over the study, and whether the authors had access to any identifying information.

5.

We note that Figures 2 and 5 in your submission contain [map/satellite] images which may be copyrighted. All PLOS content is published under the Creative Commons Attribution License (CC BY 4.0), which means that the manuscript, images, and Supporting Information files will be freely available online, and any third party is permitted to access, download, copy, distribute, and use these materials in any way, even commercially, with proper attribution. For these reasons, we cannot publish previously copyrighted maps or satellite images created using proprietary data, such as Google software (Google Maps, Street View, and Earth). For more information, see our copyright guidelines: http://journals.plos.org/plosone/s/licenses-and-copyright.

1.    You may seek permission from the original copyright holder of Figure(s) [#] to publish the content specifically under the CC BY 4.0 license. 

Reviewers' comments:

Reviewer's Responses to Questions

**Comments to the Author**

1. Is the manuscript technically sound, and do the data support the conclusions?

Reviewer #1: Yes

Reviewer #2: Yes

2. Has the statistical analysis been performed appropriately and rigorously? 

Reviewer #1: Yes

Reviewer #2: Yes

3. Have the authors made all data underlying the findings in their manuscript fully available?

Reviewer #1: Yes

Reviewer #2: Yes

4. Is the manuscript presented in an intelligible fashion and written in standard English?

Reviewer #1: Yes

Reviewer #2: Yes

5. Review Comments to the Author

Reviewer #1: This paper analyzes a relevant social issue, which is migration under humanitarian crisis. It is a valuable contribution from the computational social science perspective, and shows that the usage of data from social media can help to real-time track migration events. The paper is clearly written and the methodology is well described, so I think that it can be a valuable contribution for PLOS ONE.

However, I have a couple of minor issues and doubts which I suggest that the authors address:

1) In Fig. 1 and Fig. 5, I think that the correlation values in term of the R^2 do not provide a clear highlight when the distribution of values seems to obbey a power law. This R^2 is probably mainly dominated by the first 2 or 3 countries. I suggest to remove this indications, or either justify their usage, or either compute them in a logarithmic scale.

2) It would be interesting to point out that this methodology does not fit well for some countries in the South Cone which are further from Venezuela, like Chile (~290k "official" migrants vs ~800k? estimated) and Argentina (~130k "official" vs ~600k? estimated). This might be related to higher Twitter penetration in those countries, which promotes its usage by migrants; or to the correlation between distance travelled and socio-economic status, or to other factors that make the upscaling work incorrectly for those countries. I think that this issue deserves to be briefly discussed.

3) I did not understand the following phrase in the Discussion: "We could have used a stricter criterion and request two or more tweets abroad but this does not affect the average flows (the upscaling factors absorb it), although it notably enhances the statistical fluctuations".

As far as I understand, the scaling factor is computed as the ratio between Venezuela population and the amount of residents (TUV's). The criterion for detecting a migration situation does not affect any of the previous quantities. In this sense, if we apply a stricter "migration criterion" then the upscaled migration amount will be affected as well.

Reviewer #2: The authors propose a novel method for assessing and studying the phenomenon of migration using twitter geo-located data. The authors apply the method to the Venezuelan Migration crisis showing they are able to estimate the amounts of migrants in certain years. The estimates are compatible with those found by international organizations. Moreover, they provide a way to study in detail the geographic distribution of routes of migration.

I find the idea of using Twitter data for migrations quite appealing, despite the limitations this kind of data might have (even though the authors provide a discussion of such limitations in the conclusions). Researchers interested in migration patterns do not have always access to private data from mobile phone companies, and surveys made by international organizations might not have the desired level of detail for certain studies.

Hence, I would recommend the article for publication with minor revisions I am certain the authors will be able to address easily:

- in page 9 the authors propose a way to estimate the fluxes crossing the border of the Venezuelan country. However, I was not able to understand precisely how this is done (maybe due to my limited comprehension ability). Is it estimated by counting the number of vectors crossing the line? Are the authors able of following the trajectory of an individual and hence assess whether he crosses the border? Please rephrase it better in the text.

- The authors at page 9 makes distinction between migration patterns on land and by airplane. Of course the second ones belong to less disadvantaged individuals but still the flux might be relevant for migration studies. Do the authors think it would be possible to identify this kind of migrations as well?

- In the discussion section the authors state that the work has proven to be helpful for humanitarian agencies. Do they mean it has already been applied by these agencies for some of their studies? In this case, I would add a reference if available. Otherwise, I would state that the method "could be useful" or "have potential use for" these agencies.

- In general the work is interesting due to the fact the data used is publicly available. I would discuss a little bit about possible comparisons with private data in order to further validate the method.

After having addressed this minor comments, in my opinion the work will be ready for publication.

6. PLOS authors have the option to publish the peer review history of their article (what does this mean?). If published, this will include your full peer review and any attached files.

Reviewer #1: No

Reviewer #2: No

---

## [Author Response · Author response to Decision Letter 0]

8 Jan 2020

Answers to reviewers' comments:

First of all, we would like to thank the reviewers for the positive and constructive comments.

Reviewer #1:

This paper analyzes a relevant social issue, which is migration under humanitarian crisis. It is a valuable contribution from the computational social science perspective, and shows that the usage of data from social media can help to real-time track migration events. The paper is clearly written and the methodology is well described, so I think that it can be a valuable contribution for PLOS ONE.

However, I have a couple of minor issues and doubts which I suggest that the authors address:

1) In Fig. 1 and Fig. 5, I think that the correlation values in term of the R^2 do not provide a clear highlight when the distribution of values seems to obey a power law. This R^2 is probably mainly dominated by the first 2 or 3 countries. I suggest to remove this indications, or either justify their usage, or either compute them in a logarithmic scale.

The log-log scale in the representation is only a matter of convenience since the data points have several orders of magnitude. Both figures are showing an identity relation, as we are comparing expected versus estimated flows. The important question is whether the points fall over the diagonal and, consequently, the analysis on the “goodness” of fit must be done linearly to have sense. We cannot expect any scaling law out of linearity because that would imply that the model is not reproducing well the numbers observed in official data and that a systematic bias has been introduced. We added in the captions of Fig. 1 and 5 a note to clarify this point.

2) It would be interesting to point out that this methodology does not fit well for some countries in the South Cone which are further from Venezuela, like Chile (~290k "official" migrants vs ~800k? estimated) and Argentina (~130k "official" vs ~600k? estimated). This might be related to higher Twitter penetration in those countries, which promotes its usage by migrants; or to the correlation between distance travelled and socio-economic status, or to other factors that make the upscaling work incorrectly for those countries. I think that

this issue deserves to be briefly discussed.

We thank the reviewer for noticing this distortion. This is indeed an important observation to make. We think that the factors acting here may be two: on one side, as the reviewer states, the upscaling factor here is probably affected by the fact that the penetration rate in these countries is different. From a social perspective, migrants moving to other countries may conform to the local culture in the way the social platforms are used. However, this change of habits take time and, in some cases, it can be even generations. On the other hand, the difference could be this high because of a miss-representation of the Venezuelan migrants from the official statistics. Note what happened in Brazil, where the official numbers from the UN organizations are quite below the ones from the Federal Police and ours (Table 2). This is why it is so important to get information from extra sources. As we see from Figure 2, the routes of migration from Venezuela extend up to Santiago and Buenos Aires. We added a paragraph mentioning this issue in the Discussion Section.

3) I did not understand the following phrase in the Discussion: "We could have used a stricter criterion and request two or more tweets abroad but this does not affect the average flows (the upscaling factors absorb it), although it notably enhances the statistical fluctuations". As far as I understand, the scaling factor is computed as the ratio between Venezuela population and the amount of residents (TUV's). The criterion for detecting a migration situation does not affect any of the previous quantities. In this sense, if we apply a stricter "migration criterion" then the upscaled migration amount will be affected as well.

As the reviewer observed it is true that “the scaling factor is computed as the ratio between Venezuela population and the amount of residents (TUV's)”. The sentence "We could have used a stricter criterion and request two or more tweets abroad but this does not affect the average flows (the upscaling factors absorb it), although it notably enhances the statistical fluctuations" is intended to say that if we want to use a stricter criterion to classify people abroad as migrants, we should be consistent on the residents classification as well. In this sense, if we want to check for two consecutive tweets abroad in a specific country, in order not to lose the consistency, we should ask for residents to tweet at least twice in that year to consider them as active. On the other hand, by doing this, one notably reduces the sampling of the data and narrows the statistics, hence we discarded this option. In order to make the text clearer, we added this reflection to the above sentence.

Reviewer #2:

The authors propose a novel method for assessing and studying the phenomenon of migration using twitter geo-located data. The authors apply the method to the Venezuelan Migration crisis showing they are able to estimate the amounts of migrants in certain years. The estimates are compatible with those found by international organizations. Moreover, they provide a way to study in detail the geographic distribution of routes of migration. I find the idea of using Twitter data for migrations quite appealing, despite the limitations this kind of data might have (even though the authors provide a discussion of such limitations in the conclusions). Researchers interested in migration patterns do not have always access to private data from mobile phone companies, and surveys made by international organizations might not have the desired level of detail for certain studies. Hence, I would recommend the article for publication with minor revisions I am certain the authors will be able to address easily:

- in page 9 the authors propose a way to estimate the fluxes crossing the border of the Venezuelan country. However, I was not able to understand precisely how this is done (maybe due to my limited comprehension ability). Is it estimated by counting the number of vectors crossing the line? Are the authors able of following the trajectory of an individual and hence assess whether he crosses the border? Please rephrase it better in the text.

We thank the reviewer for pointing out this issue on the understanding of what are the numbers estimated in our methods. In the previous part of the manuscript, the method used to define a previously classified Venezuelan resident as a migrant was to detect at least one tweet from them in a second country as a proof of border crossing (Fig.1 and Table 2). We added this explanation in Validation of external flows subsection, lines 246 and following. On the other hand, from line 291, we introduce a different method, which requires a stricter sampling. We now want to assess whether migrants moving on the ground crossed a specific line along their trajectory. The results of this new measure are depicted in Figure 3 and Table 3. In order to be sure that they crossed the dashed line, we have to take at least one tweet on one side and one tweet on the other side of the dashed line. The vectorial depiction is a way to characterize the general direction of movement but it is not used to count the number of crossings. We added a clarification regarding this measure in lines 293 and following.

- The authors at page 9 makes distinction between migration patterns on land and by airplane. Of course the second ones belong to less disadvantaged individuals but still the flux might be relevant for migration studies. Do the authors think it would be possible to identify this kind of migrations as well?

There is possibility to detect air trips by having tweets of the same user in two faraway places and with a time interval compatible with a flight speed (between 300 and 900 km/h). We found a few cases in our data but they are not enough to do proper statistics. One can always use more relaxed criteria, like assuming that tweets happening between distant locations are footprints of air displacements regardless of the time between them. However, this can lead to false positives, like people who traveled on the ground by car/bus and never tweeted along the route.

- In the discussion section the authors state that the work has proven to be helpful for humanitarian agencies. Do they mean it has already been applied by these agencies for some of their studies? In this case, I would add a reference if available. Otherwise, I would state that the method "could be useful" or "have potential use for" these agencies.

It must be noticed that part of the authors belongs to UNICEF, specifically they are based in Brasilia and New York. Some of them are operatives and our results and data were discussed during the decision-making process regarding the Venezuelan crisis in Brazil and other nearby countries. This statement was included by them during the writing process and the rest of authors has no reason to consider it as false.

In particular, the insights from this research helped UNICEF to keep a broader vision of the scale of the migration problem beyond the border with Venezuela, which was the case before. Based on this, the UNICEF team moved into looking at (i) how to integrate the humanitarian response into our regular program of cooperation, especially our Municipal Seal of Approval; and (ii) expanding the reach of an AI-inspired project on xenophobia beyond the State of Roraima, close to the border.

- In general the work is interesting due to the fact the data used is publicly available. I would discuss a little bit about possible comparisons with private data in order to further validate the method.

We have added a paragraph in the Discussion section commenting on the possible comparisons that one could make with other data sources like private data.

After having addressed this minor comments, in my opinion the work will be ready for publication.

---

## [Decision Letter · Decision Letter 1]

26 Feb 2020

Migrant mobility flows characterized with digital data

PONE-D-19-28022R1

Dear Dr. MAZZOLI,

We are pleased to inform you that your manuscript has been judged scientifically suitable for publication and will be formally accepted for publication once it complies with all outstanding technical requirements.

With kind regards,

Jordi Paniagua

Academic Editor

PLOS ONE

Additional Editor Comments (optional):

Reviewers' comments:

Reviewer's Responses to Questions

**Comments to the Author**

1. If the authors have adequately addressed your comments raised in a previous round of review and you feel that this manuscript is now acceptable for publication, you may indicate that here to bypass the “Comments to the Author” section, enter your conflict of interest statement in the “Confidential to Editor” section, and submit your "Accept" recommendation.

Reviewer #2: All comments have been addressed

2. Is the manuscript technically sound, and do the data support the conclusions?

Reviewer #2: Yes

3. Has the statistical analysis been performed appropriately and rigorously? 

Reviewer #2: Yes

4. Have the authors made all data underlying the findings in their manuscript fully available?

Reviewer #2: Yes

5. Is the manuscript presented in an intelligible fashion and written in standard English?

Reviewer #2: Yes

6. Review Comments to the Author

Reviewer #2: All my previous comments have been sufficiently addressed. Therefore, I recommend this article for publication.

7. PLOS authors have the option to publish the peer review history of their article (what does this mean?). If published, this will include your full peer review and any attached files.

Reviewer #2: No

---

## [Editor Report · Acceptance letter]

6 Mar 2020

PONE-D-19-28022R1

Migrant mobility flows characterized with digital data

Dear Dr. Mazzoli:

I am pleased to inform you that your manuscript has been deemed suitable for publication in PLOS ONE. Congratulations! Your manuscript is now with our production department.

With kind regards,

on behalf of

Dr. Jordi Paniagua

Academic Editor

PLOS ONE